# *Aspergillus* Galactomannan Titer as a Diagnostic Marker of Invasive Pulmonary Aspergillosis in Lung Transplant Recipients: A Single-Center Retrospective Cohort Study

**DOI:** 10.3390/jof9050527

**Published:** 2023-04-28

**Authors:** Eun-Young Kim, Seung-Hyun Yong, Min-Dong Sung, A-La Woo, Young-Mok Park, Ha-Eun Kim, Su-Jin Jung, Song-Yee Kim, Jin-Gu Lee, Young-Sam Kim, Hyo-Chae Paik, Moo-Suk Park

**Affiliations:** 1Division of Pulmonary and Critical Care Medicine, Department of Internal Medicine, Severance Hospital, Yonsei University College of Medicine, 50-1, Yonsei-ro, Seodaemun-gu, Seoul 03722, Republic of Korea; 2Department of Thoracic and Cardiovascular Surgery, Severance Hospital, Yonsei University College of Medicine, 50-1, Yonsei-ro, Seodaemun-gu, Seoul 03722, Republic of Korea; 3Division of Infectious Disease, Department of Internal Medicine, Severance Hospital, Yonsei University College of Medicine, 50-1, Yonsei-ro, Seodaemun-gu, Seoul 03722, Republic of Korea

**Keywords:** invasive pulmonary aspergillosis, *Aspergillus* galactomannan antigen, lung transplantation

## Abstract

Invasive pulmonary aspergillosis (IPA) can occur in immunocompromised patients, and an early detection and intensive treatment are crucial. We sought to determine the potential of Aspergillus galactomannan antigen titer (AGT) in serum and bronchoalveolar lavage fluid (BALF) and serum titers of beta-D-glucan (BDG) to predict IPA in lung transplantation recipients, as opposed to pneumonia unrelated to IPA. We retrospectively reviewed the medical records of 192 lung transplant recipients. Overall, 26 recipients had been diagnosed with proven IPA, 40 recipients with probable IPA, and 75 recipients with pneumonia unrelated to IPA. We analyzed AGT levels in IPA and non-IPA pneumonia patients and used ROC curves to determine the diagnostic cutoff value. The Serum AGT cutoff value was 0.560 (index level), with a sensitivity of 50%, specificity of 91%, and AUC of 0.724, and the BALF AGT cutoff value was 0.600, with a sensitivity of 85%, specificity of 85%, and AUC of 0.895. Revised EORTC suggests a diagnostic cutoff value of 1.0 in both serum and BALF AGT when IPA is highly suspicious. In our group, serum AGT of 1.0 showed a sensitivity of 27% and a specificity of 97%, and BALF AGT of 1.0 showed a sensitivity of 60% and a specificity of 95%. The result suggested that a lower cutoff could be beneficial in the lung transplant group. In multivariable analysis, serum and BALF AGT, with a minimal correlation between the two, showed a correlation with a history of diabetes mellitus.

## 1. Introduction

Invasive fungal diseases (IFDs) are usually rare among all lung infections but are known to be more frequent in transplant recipients receiving immunosuppressants. [1] Among IFDs occurring in solid organ transplant recipients, *Candida* infection is the most common, followed by invasive aspergillosis (IA) and cryptococcosis. However, for lung transplant recipients, IA is the most common and critical form of IFDs. [2,3,4] Infection associated with *Aspergillus* commonly develops within a mean of 120 days after lung transplantation in 3–14% of patients. IA has several clinical manifestations; approximately one-third manifest as invasive pulmonary aspergillosis (IPA) [5].

*Aspergillus* fungi, which cause IPA, are ubiquitous, and transplant recipients can be easily acquired from the environment by inhalation. Invasive aspergillosis can also be a reactivation of a pre-existing focus [6]. The International Society for Heart and Lung Transplantation (ISHLT) defines IFDs as the presence of fungus in the respiratory secretions (sputum or BALF) detected by the culture, PCR, or biomarker (e.g., *Aspergillus* galactomannan antigen titer, cryptococcal antigen) in the presence of symptoms, radiologic, and endobronchial changes, or the presence of histologic changes consistent with the fungal invasion of the tissue. If there was only the presence of fungus in the respiratory tract without symptoms, radiologic, and endobronchial changes, it was considered colonization [7].

*Aspergillus* galactomannan is a polysaccharide present in the cell wall of *Aspergillus* species. The *Aspergillus* galactomannan antigen titer (AGT) test is designed to detect this polysaccharide [8]. In patients with impaired immunity, antigen testing is more specific because there are many cases with no *Aspergillus* galactomannan antibody response, even if there is invasive aspergillosis [9]. The AGT test is commercially available in both serum and bronchoalveolar lavage fluid (BALF). Depending on the patient’s situation, the clinician can choose which source to use for diagnosis. Another marker for detecting fungal infection, 1,3-beta-D-glucan (BDG), was introduced recently. BDG is a cell wall component of many fungal species, except *Cryptococcus* spp., zygomycetes, and *Blastomyces dermatitidis*, which either lack glucan entirely or produce it at a minimal level. A key advantage of using BDG analyses is that only serum is required. Serum is usually an easily accessible specimen, and due to this innate advantage, BDG analysis has been shown to significantly enhance the diagnosis of fungal infections in suspected IFD patients [10]. The development of an indirect yet simple test for detecting IPA has led to active discussion worldwide.

In the revised definition of invasive fungal disease from the European Organization for Research and Treatment of Cancer/Mycoses Study Group (EORTC/MSG) [11], cases of invasive fungal infections, including IPA, can be classified into proven, probable, and possible groups. Proven IPA diagnoses are based on a histopathologic, cytopathologic, or direct microscopic examination of a specimen, usually obtained by needle aspiration or the biopsy of infected tissue. The culture of sterile material and Aspergillus polymerase chain reaction (PCR) are also criteria for proven IPA. On the other hand, probable IPA can be defined when one or more factors are satisfied in host factor, clinical feature, and mycological evidence, respectively. From this aspect, serum AGT, BALF AGT, and BDG tests were introduced as potential indirect diagnostic tools for fungal infections. The roles of testing based on AGT, new *Aspergillus* PCR techniques, and BDG were emphasized in the meta-analysis guidelines compiled by the American Thoracic Society (ATS) in 2019 [12]. According to the meta-analysis guidelines, serum AGT tests in patients with impaired immunity suspected of having IPA had a sensitivity of 0.71 (95% confidence interval (CI), 0.64–0.78) and specificity of 0.89 (95% CI, 0.84–0.92). Similarly, BALF AGT tests in patients with impaired immunity suspected of having IPA had a sensitivity of 0.84 (95% CI, 0.73–0.91) and specificity of 0.88 (95% CI, 0.81–0.91). In this study, BDG testing has only been introduced to diagnose *Candida* infections in patients in intensive care units. The 2020 EORTC/MSG and 2019 ATS guidelines emphasize the importance of tests to differentiate indirect fungal infections [11,12,13].

Despite worldwide interest in using IPA markers in immunocompromised patients for early treatment, the role of markers distinguishing IPA from patients suffering from pneumonia of an unspecified origin is unclear, especially in lung transplantation recipients using a prophylactic antifungal agent. Additionally, there has been no research on validating serum/BALF AGT and serum BDG in diagnosing IPA in lung transplant recipients in Korea. Therefore, we investigated the efficacy of IPA markers in a group of lung transplant recipients to determine their usefulness as predictors of IPA.

## 2. Materials and Methods

This study is of a retrospective design, and patients who underwent lung transplantation at Severance Hospital between October 2012 and June 2018 were enrolled. Basic patient information collected from the electronic medical record (EMR), radiologist reports of a chest CT image, serum, BALF, and tissue biopsy results were analyzed (Figure 1). Initially, 192 recipients were enrolled in the study. The inclusion and exclusion criteria were as follows:

### 2.1. Inclusion and Exclusion Criteria

Recipients with chest CT scan records and diagnostic markers of IPA, including serum and BALF AGT tests, and with at least one or more pneumonia events during the follow-up period, were included. Recipients without documented pneumonia by chest CT scans, or for whom no diagnostic markers were available, were excluded. After data collection, the patients were classified into three groups: those having either proven or probable IPA or pneumonia unrelated to IPA. If imaging and IPA markers were collected within one week, the case was considered a single pneumonia event. If diagnostic markers were collected several times within a week of the date the chest CT was performed, the highest value was selected and used for the analysis. We defined proven and probable IPA cases following the EORTC/MSG definition [11,13]. The briefly summarized criteria are as follows.

### 2.2. Proven IPA

In this study, proven IPA was defined as a case in which the presence of the fungus was confirmed by a histopathologic, cytopathologic, or direct microscopic examination of specimens, including lung parenchymal tissue obtained via bronchoscopic tissue biopsy. No other sterile tissue samples were available, and *Aspergillus* PCR is unavailable in our center.

### 2.3. Probable IPA

Because all subjects were taking immunosuppressants, they were considered to meet the host factor of immunocompromised status, according to EORTC guidelines. Thus, patients with conditions that met both clinical criteria (e.g., suspicious fungal infection on CT, tracheobronchitis found upon bronchoscopy) and mycological criteria (including positive results from either serum or BALF AGT, growth of aspergillus in sputum/BALF culture) were defined as probable IPA. Because of the time interval between the two different factors, a case with clinical and mycological factors found within a week of one another was considered a single event of probable IPA.

### 2.4. Pneumonia Unrelated to IPA

Recipients who suspected pneumonia but did not meet the criteria for either proven or probable IPA and had no clinical evidence of fungal pneumonia were defined as having pneumonia unrelated to IPA (non-IPA pneumonia) and were placed in that group. Elevated serum and BALF AGT and serum BDG levels were not considered for exclusion, but if there was evidence of *Aspergillus* in the BALF or sputum culture, these cases were excluded.

If a recipient was repeatedly diagnosed with IPA or non-IPA pneumonia during the follow-up phase of the enrollment period, each event was included in the study and considered to be an individual case. In the same patient, individual events were defined as events that were separated by at least a month. To minimize errors deriving from recipients having events of multiple categories, all patients were enrolled as a proven IPA study group when a proven IPA event and a probable IPA event occurred simultaneously. However, there was only one case in which this occurred. In this study, we excluded cases that fell within the “possible IPA” category defined by EORTC/MSG [11,13] because of the relatively low predictive value for fungal infection.

### 2.5. Antifungal Prophylaxis and Treatment

Fungal prophylaxis is commonly recommended in lung transplantation recipients. Ideal antifungal prophylaxis strategies for lung transplant recipients have yet to reach a consensus. There are three strategies, according to ISHLT definitions, which are universal prophylaxis, targeted prophylaxis, and preemptive therapy. Universal prophylaxis refers to an antifungal medication initiated in the postoperative period in all patients before any post-transplant isolation of a fungal pathogen. Targeted prophylaxis refers to an antifungal medication initiated in the post-transplant period before fungal pathogens are isolated or the serological markers of fungus are positive in patients at high risk for IFDs (e.g., single lung transplant, early airway ischemia, rejection or change in immunosuppression, pre-transplant colonization). Preemptive antifungal therapy is an antifungal medication strategy initiated in the post-transplant isolation of a fungal pathogen or serologic marker of fungus without any evidence of an invasive fungal infection [7,14].

When IA was diagnosed, voriconazole was recommended as the drug of choice by the Infectious Diseases Society of America (IDSA) and the American Society of Transplantation (AST) [15,16]. Another therapeutic option is isavuconazole, which has been identified as non-inferior to voriconazole in invasive mold infections caused by *Aspergillus* and other filamentous fungi [17]. Posaconazole is used in refractory or infection intolerant to other first-line antifungal agents [18].

In our center, the protocol for the management of lung transplantation recipients was established by the multidisciplinary team. Per protocol, itraconazole (suspension 200 mg/20 mL twice daily) was used for prophylactic purposes for every recipient during the initial six months after lung transplantation (universal prophylaxis). Furthermore, recipients with probable or proven IPAs received voriconazole as the initial antifungal treatment [19,20].

### 2.6. Statistical Analysis

To evaluate the predictive value of tests based on AGT and serum BDG, we used the sensitivity (%), specificity (%), accuracy (%), and area under the receiver operating characteristic curve (ROC) of the sensitivity as plotted on a specificity graph. Univariate and multivariate logistic regression models were used to calculate the odd ratio for parameters related to the diagnosis of IPA. Standard definitions were used to calculate diagnostic statistics. All statistical analyses were performed using SPSS version 26 software (IBM Corp., Armonk, NY, USA) and R 4.0.2 (R Foundation for Statistical Computing, Vienna, Austria).

## 3. Results

Fifty-one recipients did not meet the inclusion criteria. In detail, 29 had no evidence of pneumonia on the chest CT, 18 did not examine fungal markers, 1 recipient had another fungal infection (Cryptococcosis), and 3 had an interval between chest CT and fungal markers of more than a week. Finally, 141 recipients were enrolled in this study. After investigating the information of all recipients, the group was classified into 26 recipients with proven IPA, 40 recipients with probable IPA, and 75 recipients with non-IPA pneumonia. After calculating the recurrent events of the individuals, we found that there were 39 cases of proven IPA, 61 cases of probable IPA, and 164 cases of pneumonia unrelated to IPA (non-IPA pneumonia) (Figure 1).

Table 1 shows the demographic data of all the recipients. The median age of the recipients was 55 (range 16–75). Male recipients were the most common (63.1%), and the median BMI was 20.6 (kg/m^2^). The most common reason for lung transplantation was idiopathic pulmonary fibrosis (49.6%). Hypertension and diabetes mellitus (DM) were lung transplant recipients’ most common underlying comorbidities. The mean level of serum AGT was 0.49 (±1.22 standard deviation (SD)), the BALF AGT was 1.46 (±1.81 SD), and the serum BDG titer was 191 (±275 SD). Among all the combined fungal infections related to IPA, *Candida* infections were the most common (52.5%), and *Saccharomyces* was the second most common source of co-infection (4.3%). Most recipients received double lung transplantation (97.9%). When the proven IPA, probable IPA, and other pneumonia groups were compared, the BALF AGT test was the only test that showed statistically significant differences across all groups (*p* < 0.001), whereas the other tests did not show statistically significant differences across the groups.

Figure 2 shows the ROC curve for the serum and BALF AGT tests and compares proven or probable IPA with non-IPA pneumonia. This study classified the proven and probable IPA as the IPA group. The results showed that the sensitivity and specificity of the serum AGT test were 50% and 91%, respectively, with an AUC of 0.724. The serum AGT cutoff value was 0.560 (Figure 2A). On the other hand, BALF AGT showed a sensitivity and specificity of 85% and 85%, respectively, and an AUC of 0.895. The cutoff value calculated from BALF AGT was 0.600 (Figure 2B).

The serum BDG values extracted from the IPA and non-IPA pneumonia groups were analyzed for their diagnostic power. However, serum BDG did not show sufficient AUC for statistical significance in both groups, with an AUC of 0.686 (sensitivity 70% and specificity 66%, respectively) (Figure 3).

Univariate and multivariate logistic regression analyses of all factors related to the diagnosis of the IPA were performed to determine the correlation between factors and IPA diagnosis (proven and probable IPA were counted as IPA diagnoses) (Table 2). Cutoff values calculated from IPA versus pneumonia unrelated to IPA were used. Serum and BALF AGT levels above the cutoff values were considered positive (serum AGT as 0.560 and BALF AGT as 0.600). A positive value was counted for BDG levels above 80 pg/mL, which denotes the cut-off value in the revised guidelines in 2020. As a result, univariate analysis showed underlying history of DM, CVD, serum AGT, and BALF AGT statistical significance, with an odds ratio (OR) of the underlying DM history of 3.45 (1.98–6.07, *p*-value < 0.001, 95% CI), underlying CVD history of 3.52 (1.76–7.33, *p* = value 0.001, 95% CI), serum AGT of 4.04 (2.34–7.80, *p*-value < 0.001, 95% CI), and BALF AGT of 7.07 (3.64–15.22, *p*-value < 0.001, 95% CI). Serum BDG did not show statistical significance (*p*-value 0.051). In multivariate analysis, the underlying history of DM, serum AGT, and BALF AGT test showed a statistical significance with an OR of underlying DM history of 3.73 (1.45–9.98, *p*-value 0.007, 95% CI), serum AGT of 3.11 (1.54–8.26, *p*-value 0.011, 95% CI), and BALF AGT of 4.80 (2.58–10.52, *p*-value < 0.001, 95% CI) (Table 2).

A correlation analysis was conducted between serum AGT and BAL AGT, assuming both markers can rise in IPA. The two markers had a weak correlation when using Pearson correlation analysis, showing a correlation coefficient of 0.296 (0.170–0.412, 95% CI, *p*-value < 0.001) (Figure 4).

## 4. Discussion

The fundamental theory of this study was that among all newly developed pneumonia in lung transplant recipients, a certain level of elevated IPA markers would distinguish true IPA from bacterial or other pneumonia. Based on this concept, comparing the serum and BALF AGT between the IPA and non-IPA pneumonia groups showed acceptable sensitivity, specificity, and ROC in diagnosing IPA, proving that it is a useful diagnostic marker.

Fungal infections can occur in organ transplant recipients, and if they are not promptly diagnosed and treated at an early stage, they can lead to fatal results. In the present study, the sensitivity of the serum AGT test was lower than that reported in previous studies. The reason for this may be the difference between the compared group (a non-lung infection group in other studies) and the small sample size. Regardless, the serum and BALF AGT test showed acceptable sensitivity and specificity, even at a very low level. The result indicates that acquiring serum and BALF AGT must be considered if IPA is strongly suspected in lung transplant recipients or other clinical evidence cannot distinguish IPA from the current event of pneumonia. However, the serum BDG test failed to show sufficient AUC to prove usefulness in this study. One reason for explaining the result is the relatively small sample size. Because BDG was introduced to our institution in 2016, only a small number of tests were conducted; thus, the sample size was small compared to other IPA markers. Additionally, the fact that serum BDG is a common metabolite of various fungal infections, not only *Aspergillus* spp., may have affected the negative result [21].

Univariate analysis revealed that serum and BALF AGT showed statistical power among the three IPA diagnostic markers except BDG. Other than the IPA markers, the history of DM and CVD were related to the diagnosis of IPA. In the multivariable analysis, a history of DM, serum AGT, and BALF AGT showed a statistical correlation with the diagnosis of IPA. This result may be associated with the high diagnostic value of the BALF AGT, considering BAL is conducted on the highly suspected site of infection directly. On the other hand, serum AGT reflects systemic circulated fungal derived, which may be affected through the individual recipient’s metabolic and excretion process. Considering the serum BDG, again, the lack of case numbers may have affected the negative result. Lastly, IPA correlation with DM can be explained by the fact that DM patients are known to have a higher incidence of infection. So, this may be one of the reasons that DM showed a correlation in both univariate and multivariate analysis in this study.

Similar results have been previously reported. According to a meta-analysis of fungal titers in lung transplant patients published by Sabine et al. in 2019, the sensitivity of the serum AGT test was reported to be 71%, and the specificity was 85%. The BALF AGT test also showed a high specificity and sensitivity. Although the BALF AGT results vary from study to study, the sensitivity was reported to be 60–100%, and the specificity was reported to be 40–95%. The serum BDG test was reported as having 71% sensitivity and specificity of 85% for serum, the BALF test had a sensitivity of 80%, and the specificity was 38.5–81.8% [22].

Prior to that, Christian et al. published research on IPA in lung transplant recipients in 2014 and the usefulness of BALF AGT tests; both the sensitivity and specificity were reported to be over 90%. The paper also stated that an evaluation of ‘how useful serum BDG tests were’ was necessary [23,24].

Another study evaluated the diagnostic accuracy of BALF AGT tests. In 2007, Shahid et al. reported a retrospective analysis of 333 BALF samples from 116 lung transplant recipients. That study used a documented cutoff value of ≥0.5 and showed that the BALF AGT test had a sensitivity of 60% and a specificity of 95% compared with recipients without IPA [24]. A previous paper has also dealt with the sensitivity and specificity of BAL AGT in a mixed population (e.g., solid-organ transplant recipients, hematologic malignancies, nonhematologic malignancies, intensive care admission) in the diagnosis of IPA [23]. They defined proven IPA and probable IPA by applying the revised EORTC/MSG consensus definitions for IA criteria, as in this study [13]. However, critically ill patients with chronic obstructive pulmonary disease, autoimmune disease, or liver cirrhosis were added as host factors. Additionally, AGT detection in BALF was excluded as a mycological criterion to avoid incorporation bias. In BALF AGT analysis, they analyzed proven IPA and probable IPA by grouping them into IPA. Patients classified with possible IPA were excluded from the analysis because of the possibilities of an ambiguous diagnosis. A total of 251 BALF samples were analyzed, and 59 cases (23.6%) were classified as IPA. The performance of AGT detection in BALF samples was calculated for different optical density (OD) cutoff indices with a 95% CI. The sensitivity–specificity analysis showed an inverse relationship. An OD index value of ≥3.0 corresponded to 100% specificity. Conversely, negative results virtually always ruled the disease out since there were very few false negatives. An ROC curve was calculated to determine the most appropriate OD index cutoff to define positivity, indicating 0.8 as the most optimal cutoff value for this mixed patient population [23].

In the EORTC/MSG guidelines revised in 2020, the cut-off values of serum AGT and BALF AGT, which indicated the probable IPA aspect of mycological criteria, were defined as 1.0 or higher, respectively. It presents a much higher value than the cut-off value of 0.560 and 0.600 for serum AGT and BALF AGT, respectively, used in this study.

Physicians commonly encounter pneumonia in lung transplantation recipients, but it is difficult to distinguish between IPA and pneumonia unrelated to IPA. Many previous studies have reported diagnostic accuracy with acceptable sensitivity and specificity of several fungal markers; however, most studies introduced previously were designed to compare a group without any signs of lung infection with a group of IPA patients to validate the diagnostic potency of the IPA makers. There are limitations in accepting the results of existing studies because they differ from the actual situation. For these reasons, diagnosing IPA and differentiating it from common bacterial pneumonia in lung transplant recipients has not been answered clearly.

In this study, we compared data from an IPA group with data from a group that had pneumonia unrelated to IPA, emphasizing the importance of fungal markers for distinguishing IPA from other types of pneumonia. In this respect, this study has several strengths. Taking advantage of this strength, it might be possible to extend this study’s design into a mixed-population model. Moreover, to the best of our knowledge, this study is the first study to validate the AGT test as a diagnostic marker for IPA in lung transplant recipients in South Korea.

However, this study had several limitations. First, the analysis of the relatively small patient group size, limited to a single center, may have influenced and limited the test results. Second, all the pneumonia events that occurred during the follow-up period for each recipient were used for the analysis. Therefore, there is a possibility that the results for each event were continuously related and may have been misdiagnosed as a false-negative group of recipients who did have IPA. Last, because of frequent serum and BALF test results in the probable IPA group, selecting the highest value may have caused biased results.

## 5. Conclusions

Both serum and BALF AGT tests were useful as predictors of IPA in lung transplantation recipients, with a relatively low sensitivity of serum AGT and superior diagnostic value of BALF AGT. The serum BDG level did not show a correlation or adequate cutoff value in the diagnosis of IPA in this study. Revised EORTC suggests a cutoff value of 1.0 in both serum and BALF AGT for the diagnosis of IPA, but our study showed that in the lung transplant population, the cutoff value should be optimized. Further studies with larger sample sizes are required to evaluate serum AGT and serum BDG levels to determine their adequacy in lung transplantation recipients.

## Figures and Tables

**Figure 1 jof-09-00527-f001:**
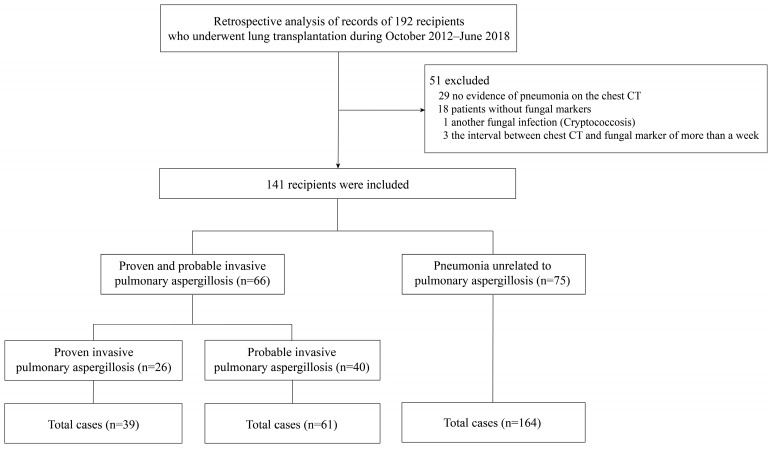
Summary of the study population.

**Figure 2 jof-09-00527-f002:**
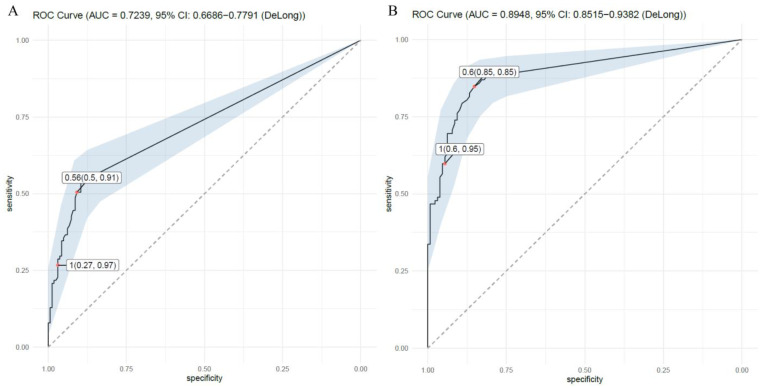
The ROC curve for serum AGT and BAL AGT of proven/probable IPA (**A**). Serum AGT cut-off for differentiating proven/probable IPA. Serum AGT of 0.560 showed a sensitivity of 50% and specificity of 91%. Serum AGT of 1.00 showed a sensitivity of 27% and specificity of 97% (**B**). BAL AGT cut-off for differentiating proven/probable IPA. BALF AGT of 0.600 showed a sensitivity of 85% and specificity of 85%. BALF AGT of 1.000 showed a sensitivity of 60% and specificity of 95%.

**Figure 3 jof-09-00527-f003:**
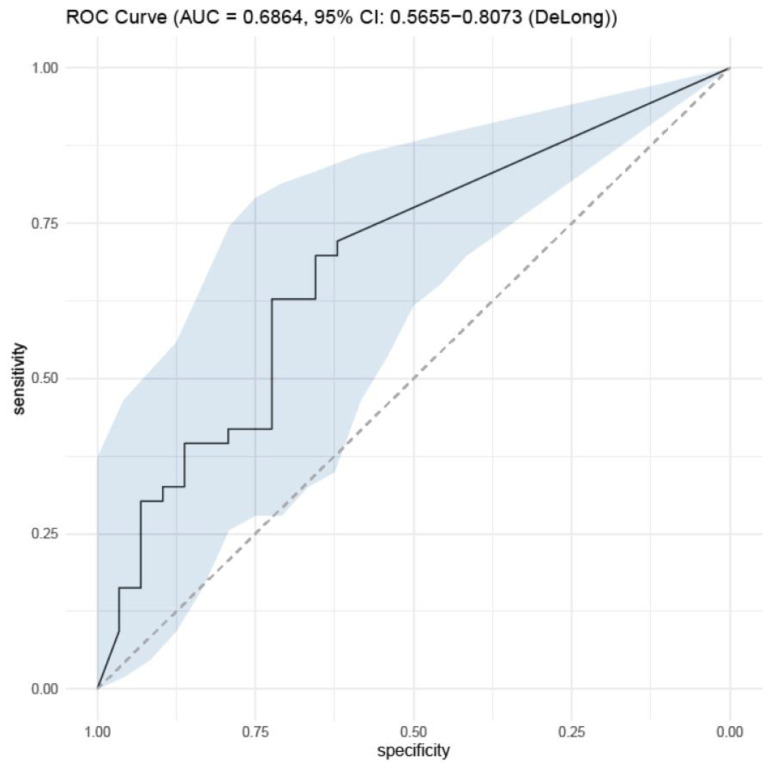
ROC curve for beta-D Glucan of proven/probable IPA.

**Figure 4 jof-09-00527-f004:**
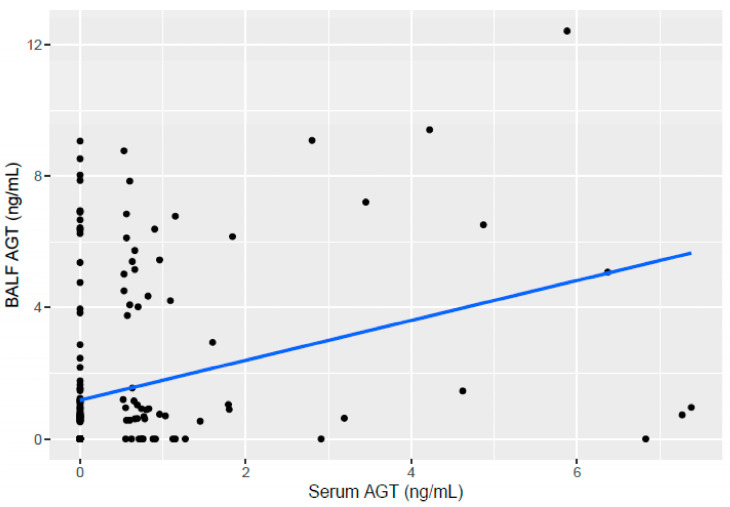
The correlation between serum AGT and BALF AGT.

**Table 1 jof-09-00527-t001:** The demographic data of all the recipients.

Demographic Variables	Proven IPA	Probable IPA	Non-IPA Pneumonia	Total
Number of patients, *n*	26	40	75	141
Total cases, *n*	39	61	164	264
Age at transplantation, years median (IQR)	55 (18–73)	54 (17–71)	55 (16–75)	55 (16–75)
Male, *n* (%)	17 (63.4)	25 (62.5)	47 (62.7)	89 (63.1)
BMI (kg/m^2^), median (IQR)	19.6 (11.6–25.2)	21.0 (13.4–32.9)	20.6 (12.9–29.5)	20.6 (11.6–32.9)
Underlying cause of transplantation, *n* (%)				
IPF	14 (53.8)	18 (45.0)	38 (50.2)	70 (49.6)
Other lung disease *	12 (46.2)	22 (55.0)	37 (49.8)	71 (50.4)
Underlying disease, *n* (%)				
Hypertension	6 (23.1)	9 (22.5)	17 (22.7)	32 (22.7)
Diabetes mellitus	11 (42.3)	10 (25.0)	12 (16.0)	33 (23.4)
Chronic kidney disease	0 (0.0)	4 (10.0)	4 (5.3)	8 (5.7)
Liver cirrhosis	0 (0.0)	0 (0.0)	1 (1.3)	1 (0.7)
CVD	3 (11.5)	4 (10.0)	8 (10.6)	15 (10.6)
Level, mean ± SD				
Serum *Aspergillus* galactomannan antigen (index)	1.12 ± 1.81	1.03 ± 171	0.13 ± 0.48	0.49 ± 1.22
BALF *Aspergillus* galactomannan antigen (index)	4.58 ± 3.31	2.24 ± 2.50	0.2 ± 0.55	1.46 ± 1.81
Serum beta-D Glucan titer (pg/mL)	204 ± 255	267 ± 320	107 ± 211	191 ± 275
Combined fungal infection, *n* (%)				
*Candida* spp.	10 (38.5)	24 (60.0)	40 (53.3)	74 (52.5)
*Mucormycosis*	0 (0.0)	0 (0.0)	0 (0.0)	0 (0.0)
*Saccharomyces* spp.	1 (3.9)	4 (10.0)	1 (1.3)	6 (4.3)
Other fungal infection	1 (3.9)	2 (5.0)	2 (2.6)	5 (3.5)
Type of lung transplant, *n* (%)				
Double	24 (92.3)	40 (100)	74 (98.7)	138 (97.9)
Single	2 (7.7)	0 (0.0)	1 (3.0)	3 (2.1)

Abbreviation: PBSCT, peripheral blood stem cell transplantation; IPF, idiopathic pulmonary fibrosis; ILD, interstitial lung disease; CVD, cardiovascular disease; BALF, bronchoalveolar lavage fluid. * Other lung diseases include bronchiolitis obliterans after PBSCT, connective tissue disease-related ILD, ILD other than IPF, and bronchiectasis.

**Table 2 jof-09-00527-t002:** Univariate and multivariate analysis of risk factors associated with a diagnosis of IPA.

	Univariate		Multivariate	
Variables	OR (95% CI)	*p*-Value	OR (95% CI)	*p*-Value
Female sex	1.04 (0.63–1.72)	0.877	2.14 (0.91–5.16)	0.084
Age (<30)				
31–40	0.26 (0.05–1.17)	0.090	0.15 (0.01–2.08)	0.169
41–50	0.38 (0.07–1.59)	0.199	0.05 (0.00–0.82)	0.042
51–60	0.22 (0.04–0.89)	0.040	0.06 (0.00–0.84)	0.042
61–70	0.37 (0.07–1.50)	0.177	0.06 (0.00–0.86)	0.044
>70	0.30 (0.04–2.08)	0.236	0.18 (0.00–5.76)	0.348
BMI (<15)				
15 ≤ BMI < 20	0.42 (0.16–1.08)	0.071		
20 ≤ BMI < 25	0.73 (0.28–1.86)	0.505		
25 ≤ BMI < 30	0.38 (0.12–1.16)	0.092		
Underlying cause of transplantation				
ILD except IPF and RA-ILD	0.37 (0.15–0.80)	0.017		
IPF	1.10 (0.67–1.80)	0.713		
COPD	1.04 (0.37–2.72)	0.944		
RA-ILD	2.02 (0.94–4.39)	0.072		
BO	1.05 (0.42–2.49)	0.916		
Others	1.09 (0.46–2.51)	0.838		
Lung transplantation type, double	0.92 (0.15–7.09)	0.930		
Underlying disease				
Hypertension	0.65 (0.36–1.15)	0.144		
Diabetes mellitus	3.45 (1.98–6.07)	<0.001	3.73 (1.45–9.98)	0.007
CVD	3.52 (1.76–7.33)	0.001		
Combined fungal infection				
*Candida* spp.	1.43 (0.87–2.37)	0.164		
Other fungal infection	7.24 (2.23–32.37)	0.003		
Serum AGT	4.04 (2.34–7.80)	<0.001	3.11 (1.54–8.26)	0.011
BALF AGT	7.07 (3.64–15.22)	<0.001	4.80 (2.58–10.52)	<0.001
Serum beta-D glucan	1.00 (1.00–1.01)	0.051		

Abbreviation: ILD, interstitial lung disease; IPF, idiopathic pulmonary fibrosis; COPD, chronic obstructive lung disease; RA-ILD, rheumatoid arthritis-associated interstitial lung disease; BO, bronchiolitis obliterans; AGT, Aspergillus galactomannan antigen titer; BALF, bronchoalveolar lavage fluid; CI, confidence interval. In multivariate analysis, the cut-off value of serum AGT (index) and BALF AGT (index) are 0.560 and 0.600, respectively.

## Data Availability

Not applicable.

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
