# Peer review of "Aspergillus Galactomannan Titer as a Diagnostic Marker of Invasive Pulmonary Aspergillosis in Lung Transplant Recipients: A Single-Center Retrospective Cohort Study"

_jof, 2023, doi:10.3390/jof9050527_

Round 1
Reviewer 1 Report
This is a single center, retrospective assessment of serum and BALF galactomannan and serum beta-d-glucan (BDG) testing in a cohort of 192 lung transplant recipients receiving universal itraconazole prophylaxis for the first 6 months after transplant, 26 of whom developed proven IPA, 40 probable IPA, and 75 pneumonia unrelated to IPA. Other similar studies have been conducted before over the past 2 decades, including studies specifically focused on lung transplant recipients, and I am not sure this study is novel enough to make it notable. There are also several problematic aspects and methodologic flaws in the analysis.
They used a serum galactomannan cutoff of 0.525, finding a sensitivity of 54%, specificity of 90%, and AUC of 0.719, and a BAL galactomannan cutoff of 0.88, with a sensitivity of 84%, specificity 94%, and AUC of 0.92. They found that the sensitivity of serum BDG was 85% and specificity 62%, with an AUC of 0.712. The authors performed univariable analysis to examine the relationship between these diagnostics and proven IPA diagnosis, finding on multivariable analysis that only BALF AGT was associated with proven IPA. This is generally within the wide ranges of diagnostic performance reported in the literature in solid organ transplant recipients, especially in lung transplant recipients.
It is not clear why the authors are using an older version of the EORTC/MSG guidelines vs the more recent ones (Donnelly JP, et al.) which have a significantly revised definition of what qualifies as a proven or probable IA infection. The analyses should be revised to take this into account, and the cutoff values assessed in this analysis harmonized with modern cutoff values. An antigen cutoff of 0.525 in serum and 0.88 in BALF seems arbitrary.
I think using the criterion of positive cultures in lung tissue to define a case of proven IPA is very problematic – the lung is not a sterile site and many lung transplant patients are colonized in their lungs – a positive culture, even from a biopsy specimen, does not necessarily indicate IPA if there is no concomitant histopathologic evidence of an angioinvasive/tissue invasive process.
Similarly, I think it is problematic that there is artificial inflation of the diagnostic performance of these fungal markers since positive results of either serum or BALF AGT and serum BDG were included to define probable IPA, so by definition a patient with a positive BAL galactomannan was considered to have IPA, and a patient with a positive serum galactomannan as well – the adjudicators of whether the case was a probable or proven IPA infection do not appear to have been blinded to these test results – classic incorporation bias. Diagnostic performance is probably also artificially inflated significantly by the fact that 145 episodes of ‘possible’ IPA by EORTC/MSG criteria were excluded – most studies evaluating the diagnostic performance of fungal markers would include these with the non-IPA pneumonia group (vs. proven/probable IPA together), as it is critically important in studies describing the performance of a diagnostic test in a study population to include all members of the study population. Selectively excluding patients with ‘possible’ IPA and only assessing diagnostic performance in patients with ‘proven’ IPA will significantly distort estimates of sensitivity and specificity of the tests in an artificial manner, and these calculations should be performed on the entire spectrum of patients with suspected pneumonia. It is not clear whether patients with an isolated positive BDG were considered to have IPA, when BDG is not an IPA-specific marker.
I think the multivariable models should be conducted separately for BAL and serum galactomannan – these are going to be collinear parameters and it does not make any sense to include them both in the same model. The outcome variable also should not be proven IPA alone, but proven/probable IPA as defined in the most recent EORTC/MSG definitions, vs. possible/no IPA.
A final minor point: In the introduction, I would not include a discussion of Aspergillus antibody assays, which are for the diagnosis mostly of ABPA and irrelevant to the diagnosis of invasive pulmonary aspergillosis.
Author Response
First of all, thank you for reviewing our research. As you have pointed out, we have changed the script according to your suggestion.
Comment 1: It is not clear why the authors are using an older version of the EORTC/MSG guidelines vs the more recent ones (Donnelly JP, et al.) which have a significantly revised definition of what qualifies as a proven or probable IA infection. The analyses should be revised to take this into account, and the cutoff values assessed in this analysis harmonized with modern cutoff values. An antigen cutoff of 0.525 in serum and 0.88 in BALF seems arbitrary.
Reply 1: Thank you for the suggestion. We totally agree with your comment. We conceptualized and began to collect data from 2018, so initially, we used outdated EORTC criteria for patient data collection. We change the old EORTC definition to the newer 2020, the most recent one. The 2020 EORTC definition suggests a cutoff value of 1.0 in both serum and BAL fluid for meeting mycological evidence in suspected aspergillosis, so we have counted an absolute value of 1.0 into our data and compared with a cutoff value calculated from the AUROC curve. Both sensitivity and specificity were compared to prove that altered cutoff value may show better diagnostic performance in real-world lung transplantation patients, especially when clinicians have to differentiate invasive pulmonary aspergillosis (IPA) from all suspected pneumonia patients. Our research aimed to show real-world, optimized cut-off value to distinguish IPA in lung transplant patients who concurrently suffered from various causes of pneumonia.
Changes in the text: We have changed the script on page 2, lines 73 to 81, and page 3, lines 121-132 for the change of IPA definition according to the most revised 2020 EORTC definition. Also, change page 7, lines 232-236, page 11, lines 345-348, and conclusion, page 11, lines 382-384.
Comment 2: I think using the criterion of positive cultures in lung tissue to define a case of proven IPA is very problematic – the lung is not a sterile site and many lung transplant patients are colonized in their lungs – a positive culture, even from a biopsy specimen, does not necessarily indicate IPA if there is no concomitant histopathologic evidence of an angioinvasive/tissue invasive process
Reply 2: Thank you for the direction. In our institution, lung biopsy specimens are either acquired from transbronchial lung biopsy (TBLB) and/or bronchoscopic trachea/bronchus biopsy. We agree with your comment that the lung itself is not a sterile site. But regarding IPA, the best way to obtain fungal pathogens for diagnosis is from the lung. Hematogenous spreading or CNS invasion rarely occurred in IPA patients. Supportively, in our patient group, there was no patient having such a problem. To minimize corruption bias from normal flora, only a TBLB-acquired lung parenchymal tissue and pathologic report with the presence of the fungal pathogen, suggesting aspergillus, was considered a sterile specimen from the lung.
Changes in the text: Changed script page 3, lines 121-125
Comment 3: 145 episodes of ‘possible’ IPA by EORTC/MSG criteria were excluded – most studies evaluating the diagnostic performance of fungal markers would include these with the non-IPA pneumonia group (vs. proven/probable IPA together), as it is critically important in studies describing the performance of a diagnostic test in a study population to include all members of the study population.
Reply 3: Once again, we appreciate your comment. As you have pointed out, we included a probable IPA group combined with a proven IPA group. We have re-analyzed the AUROC curve regarding change of data, univariate analysis, and multivariate analysis.
Changes in the text: Changed text on page 5, lines 188-196, Figure 1, page 6, line 217 to page 7, line 236. Change discussion on page 10, lines 282-284, 295-299. Page 11, lines 382-384.
Comment 4: It is not clear whether patients with an isolated positive BDG were considered to have IPA when BDG is not an IPA-specific marker.
Reply 4: Beta-D glucan is a universal marker for the fungal organism, except for a few species, such as cryptococcus. We agree with your comment, so our point of view for BDG was not to prove it as IPA specific marker but as an addictive marker with all other IPA-specific markers. Because of a lack of cases, this research did not show any diagnostic or addictive value. However, in the real-world BDG is more easily accessible compared to Aspergillus titer, especially considering the time of ‘sampling to result.’ BDG may be used for early detection of IPA, so in our opinion, further study is required in lung transplant patients to prove the usefulness of the BDG.
Changes in the text: There are no changes regarding this reviewer’s comment.
Comment 5: multivariable models should be conducted separately for BAL and serum galactomannan – these are going to be collinear parameters, and it does not make any sense to include them both in the same model. The outcome variable also should not be proven IPA alone but proven/probable IPA as defined in the most recent EORTC/MSG definitions vs. possible/no IPA.
Reply 5: We appreciate your comment. We conducted BAL and serum AGT separately for the multivariable model and changed the script. The proven/probable IPA was defined using the 2020 EORTC/MSG definition compared to the non-IPA group. Additionally, we plotted Pearson correlation analysis between serum and BALF AGT. The result showed a low correlation of 0.296.
Changes in the text: Changed script from page 8, Table 2, page 9, lines 262-265. Page 10, lines 305-306.
Comment 6: In the introduction, I would not include a discussion of Aspergillus antibody assays, which are for the diagnosis mostly of ABPA and irrelevant to the diagnosis of invasive pulmonary aspergillosis.
Reply 6: Thank you for your comment. As you have commented, we have changed the script according to your recommendation.
Changes in the text: Deleted whole sentence.
Reviewer 2 Report
Amongst the Proven IPA category patients had either a positive lung biopsy specimen or a blood culture; numbers for each would be interesting, I would expect the latter would be uncommon. Were serum AGT measurements carried out only on clinical suspicion of IPA or were periodic screening tests done? Was further speciation carried out on the Aspergillus isolates, was sensitivity testing done on any? Have practices been changed in the Severance Hospital unit as a result of this study?
Author Response
Comment 1: Amongst the Proven IPA category, patients had either a positive lung biopsy specimen or a blood culture; numbers for each would be interesting. I would expect the latter would be uncommon.
Reply 1: Thank you for your comments. We apologize for causing confusion. A misunderstanding was created in the process of mentioning the general diagnostic criteria presented in the guidelines in the text. As a result of checking all our data, only cases confirmed by histopathologic diagnosis in lung tissue were classified as proven IPA in this study.
Changes in the text: Changed script page 3, lines 121-125.
Comment 2: Were serum AGT measurements carried out only on clinical suspicion of IPA, or were periodic screening tests done?
Reply 2: Once again, thank you for your comments. In our institution, Lung transplant recipients regularly receive screening tests, including fungal titer. However, on the clinician’s suspicion, a CT scan and a more invasive procedure, including bronchoscopic approaches, may be considered to diagnose IPA.
Changes in the text: There are no changes regarding this reviewer’s comment.
Comment 3: Was further speciation carried out on the Aspergillus isolates? Was sensitivity testing done on any?
Reply 3: We appreciate your comments. Unfortunately, our institution routinely performed the test for identification only. The test for drug susceptibilities of Aspergillus spp. done for research purposes. It is performed only for limited purposes in cases where the patient group using voriconazole for IPA does not respond to treatment. None of our patients had an identification or drug sensitivity test for Aspergillus in our study.
Changes in the text: There are no changes regarding this reviewer’s comment.
Comment 4: Have practices been changed in the Severance Hospital unit as a result of this study?
Reply 4: Regarding your comments, we are optimizing the diagnostic value of fungal titer, the frequency of the testing, etc. Our goal is to detect IPA as early as possible. We are still in the process of our study to make a milestone for that purpose.
Changes in the text: There are no changes regarding this reviewer’s comment.